# Superhydrophobic Soot Coated Quartz Crystal Microbalances: A Novel Platform for Human Spermatozoa Quality Assessment

**DOI:** 10.3390/s19010123

**Published:** 2019-01-02

**Authors:** Karekin D. Esmeryan, Rumiana R. Ganeva, Georgi S. Stamenov, Todor A. Chaushev

**Affiliations:** 1Acoustoelectronics Laboratory, Georgi Nadjakov Institute of Solid State Physics, Bulgarian Academy of Sciences, 72, Tzarigradsko Chaussee Blvd., 1784 Sofia, Bulgaria; 2Research Department, Nadezhda Women’s Health Hospital, 3 “BlagaVest” Street, 1330 Sofia, Bulgaria; rum.ganeva@gmail.com (R.R.G.); georgistamenov@abv.bg (G.S.S.); todorangelovchaushev@gmail.com (T.A.C.)

**Keywords:** human spermatozoa, quartz crystal microbalance, soot, superhydrophobicity

## Abstract

The functionality of human spermatozoa is a key factor for the success rate of natural human reproduction, but unfortunately the infertility progressively increases due to multifarious environmental factors. Such disquieting statistics requires the employment of sophisticated computer-assisted methods for semen quality analysis, whose precision, however, is unreliable in cases of patients with low sperm concentrations. In this study, we report a novel quartz crystal microbalance (QCM) based biosensor for in-situ quality assessment of male gametes, comprising a superhydrophobic soot coating as an interface sensing material. The soot deposition on the surface of a 5 MHz QCM eliminates the noise that normally arises upon immersion of the uncoated sensor in the test liquid environment, allowing the detection of human spermatozoa down to 1000–100,000 units/mL (1–100 ppb). Furthermore, the soot coated QCM delimitates in a highly repeatable way the immotile and motile sperm cells by inducing fundamentally distinct responses in respect to sensor sensitivity and signal trends. The obtained results reveal the strong potential of the superhydrophobic QCM for future inclusion in diverse laboratory analyses closely related to the in vitro fertilization procedures, with a final aim of gaining practical approaches for diagnoses and selection of male gametes.

## 1. Introduction

Male reproductive dysfunction, also known as male infertility, is a multifactorial pathological condition that has been recognized as one of the major and persistent medical problems towards the natural evolution of the human race [1,2]. According to literature data, the male infertility affects about 7–12% of the sexually active male population and its genetic landscape is highly complex due to the involvement in spermatogenesis of more than 2000 genes with extremely heterogeneous phenotypes [1,3]. Furthermore, the reproductivity function in men is highly sensitive to and adversely affected by various environmental toxicants, including pesticides in food, alcohol consumption, drug use, etc., [4,5].

The increasingly worrying tendency of declined fertility within the past few decades have demanded the development of assisted-reproductive technologies such as the in vitro fertilization (IVF) [5,6,7]. During the IVF procedure, a female oocyte is fertilized by the spermatozoa outside the human body and it is implanted in the woman’s uterus in order to achieve clinical pregnancy [6]. The success rate of IVF strongly depends on the quality of spermatozoa, since it directly influences the process of embryogenesis and the embryo’s implantation potential [8,9]. Therefore, a mandatory and unavoidable step prior to performing artificial insemination is the semen quality assessment, which requires the accurate estimation of the spermatozoa concentration in the seminal plasma and their motility (progressive, non-progressive, or lack of motility) [10]. 

Most commonly used and modern approaches for semen analysis are based on computer-assisted (CASA) methods [11], haemocytometry [12], or flow cytometry [13]. Each of these techniques is very efficient, but regrettably there are still a few shortcomings that introduce uncertainties in defining the sperm quality correctly. For instance, the accuracy and sensitivity of a CASA system is determined by the sperm concentration (high and low concentrations cause errors), suspending medium, sample chamber depth, hardware and instrument settings, and can vary from sample-to-sample [11]. Also, the haemocytometer yields large discrepancies in terms of sperm concentrations for samples with good motility (˃50% progression) [12], while the flow cytometry suffers from difficulties associated with the special flattened shape of human sperm cells and tightly packed nature of sperm DNA [13]. Last but not least, the above discussed methods require expensive equipment and complicated setup for optimal performance. 

A powerful and low-cost, an alternative system for human spermatozoa quality assessment could comprise a quartz crystal microbalance (QCM) based analytical apparatus, often referred in the literature as thickness shear mode resonator and/or acoustic wave sensor [14]. This device correlates the changes in its resonance frequency *f_r_* and dissipation factor *D* with the mass of the attached object [15] and/or viscosity-density product of the substance if the operating environment is liquid [16]. As a result, the QCM has demonstrated capabilities for high-sensitive detection of fibrinogen in plasma [17], salt concentrations in solutions [18], residual plasmin activity in milk [19], breast cancer [20], ractopamine in feed products [21], or the anti-biofouling potential of functional coatings [22,23]. Moreover, due to the absolute sensitivity of ng/cm^2^ and preferential attachment of animal gametes on gold and Poly-l-Lysine coated surfaces, a few 5–10 MHz QCMs with gold electrode structure have recently been used for semen quality analysis of boars [24,25,26]. However, a significant disadvantage of the described QCM sensing platform is the excessive noise that is generated by the Tri X-cell diluent for sperm preservation, leading to concealed sensor signal (that associated with the arrival/attachment of immotile/motile spermatozoa) [26]. 

In this paper, we report for the first time the potential applicability of superhydrophobic carbon soot coatings in QCM based biosensors for the accurate determination of two basic human sperm characteristics (concentration and motility). By comparing and analyzing the real-time sensor response of a set of uncoated and soot coated 5 MHz QCMs, we show that the proposed superhydrophobic device configuration can detect spermatozoa concentrations as low as 1000–100,000 units/mL (0.001–0.1 µg/mL = 1–100 ppb) and clearly indicate whether the cells are motile or immotile. The major advantage of our method is the opportunity to minimize substantially the influence of viscosity of the buffer solution (i.e., the diluent), thus suppressing the unacceptable noise levels and extracting only the useful signal due to the sperm cells. 

## 2. Theoretical Aspects of the QCM’s Operation in Liquid Environments

The resonance behavior of the QCM in liquids is mainly governed by the coupling of the elastic shear wave in the crystal and the viscous shear wave in the liquid, resulting in frequency shifts that are proportional to the viscosity *η* and density *ρ* of the liquid [16]:
(1)$$\Delta f=\;-f^{\frac{3}{2}}\sqrt{\frac{\rho \eta }{\pi \mu_{q}\rho_{q}}}$$
where *πµ_q_ρ_q_* is the acoustic impedance of the quartz crystal. Such a viscous coupling also induces damping of the resonance oscillations, since there is mismatch between the acoustic impedances of the crystal and the liquid (e.g., V_quartz_ ~3000 m/s; V_H2O_ ~1500 m/s), and a certain portion of the propagating acoustic energy is lost (for the case of piezoresonance devices, the terms acoustic and electromechanical are interchangeable, since the wave has electromechanical origin, but its velocity is identical to that of the sound in solids). During the QCM operation in liquids, the damping is manifested as increase in the series (dynamic) resistance *R* and a separate study has derived a linear relationship between *R* and *ηρ* [27]: (2)$$\Delta R=\left(\frac{n\omega L}{\pi }\right).\left(\sqrt{\frac{2\omega \rho \eta }{{µ}_{q}\;\rho_{q}}}\right)$$
where *n*—number of sides in contact with the liquid, *ω*—angular frequency at series resonance, and *L*—inductance at unloaded (dry) sensor. It is pertinent to mention that some authors prefer to consider the dissipation factor *D* instead of the dynamic resistance *R* [22], but these two parameters are interrelated: (3)$$D=\frac{1}{Q}=\frac{R}{\omega L}=\omega CR$$
where *Q*—quality factor of the resonator, *C*—capacitance at unloaded (dry) sensor. In addition, the shear-horizontal displacements decay exponentially, so the oscillations perturb a small fraction of the contacting liquid medium and the QCM senses mass and/or viscosity-density changes within the shear wave penetration depth:
(4)$$\delta =\sqrt{\frac{\eta }{\pi f\rho }}$$
which at 5 MHz fundamental frequency and water-based liquids is *δ* ~250 nm. However, it has previously been demonstrated that, along with the transverse component, the QCM may launch surface normal compressional waves capable of penetrating into the bulk of the liquid and even being reflected from the upper boundary i.e., the liquid-air interface [26,28]. Hence, the reflected waves can cause the formation of cyclic compressional resonances depending on the crystal-liquid-air distance and insert significant measurement errors [26]. 

The above considerations are strictly valid for a no-slip boundary condition, which is fulfilled for hydrophilic surfaces [29]. Completely different scenarios occur upon wettability-driven molecular slip at the solid-liquid interface [30,31]. In this situation, the flow motion of molecules within a liquid, under the applied electromechanical vibrations of the QCM, depends on the solid-liquid interfacial energy *γ_sl_* that controls the energy utilized for vacancies formation *Wij*: (5)$$W_{ij}=S\gamma_{sl}\left(\cos\theta +1\right)$$

Equation (5) shows direct relationship between *Wij*, *γ_sl_* and the liquid contact angle *θ*, meaning the amount of energy reaching the liquid and being dissipated by it can be manipulated through alteration of the sensor surface’s wettability. In fact, for superhydrophobic surfaces (coatings), the air trapped within the surface asperities (known as plastron) localizes the shear-horizontal waves mostly in the crystal’s bulk [29]. In turn, the surface oscillations are effectively “decoupled” from the liquid and the generated frequency and resistance fluctuations are much lower in comparison with the reference values predicted by Equations (1) and (2) [23,29,32,33,34,35]. 

Finally, the superhydrophobic QCM configuration has previously been used to develop a hypothesis that any binding events (e.g., spermatozoa attachments) at the solid-liquid interface may trigger Cassie-Baxter to Wenzel wetting state transitions, and vice versa, allowing the detection of various biological compounds/objects via their effect on the surface wettability rather than the conventional mass loading or viscosity-density changes [36]. As a consequence, the dynamics of a wide range of antigen-antibody, nucleic acid hybridization, or receptor-ligand interactions could be monitored, even if the mass of the target analyte is below the QCM’s detection limit [36]. 

## 3. Materials and Methods

### 3.1. Human Spermatozoa Collection, Processing and Experimental Treatments

Human spermatozoa samples were collected from healthy donors with normozoospermia by masturbation after 2–5 days of sexual abstinence, in compliance with the standard procedures of the World Health Organization. Subsequently, the spermatozoa were placed in a sterile container and liquefied in an incubator for 30 min (Galaxy 170R, Eppendorf, Hamburg, Germany). Then, the sperm volume and pH were analyzed and the samples underwent manual and computer-assisted analysis of concentration, motility, and morphology. For the manual assessment, sperm with volume ≤1.5 mL was evenly distributed on a 1 × 1 mm square grid with 100 small squares in the center of a cover glass mounted in a counting chamber. The number of spermatozoa in 10 squares showed the sperm concentration (millions per milliliter), while their motility was evaluated by counting the motile and immotile cells, and calculated as percentage of the total concentration. Later, the probes were centrifuged and the residual supernatant was discarded. After sperm processing by “swim up” procedure, only the actively motile spermatozoa (those with progressive motility and curvilinear velocity of category *a* and *b*, i.e., >25 µm/s or 5–25 µm/s) were diluted in Global for fertilization medium (LifeGlobal Group, Guilford, CT, USA), set in sterile tubes, and kept readily available in an incubator at constant ambient temperature and CO_2_ content of ~37 °C and 5%, respectively. Global for fertilization is a water-based liquid used for seminal preparation and preservation prior to IVF, since its multicomponent composition shown in Table 1 meets the stringent metabolic requirements of the human sperm [37].

### 3.2. Soot Synthesis and Deposition

The soot coatings, deposited as an interface material on 5 MHz QCMs (SRS, San Diego, CA, USA), were synthesized during the controlled combustion of rapeseed oil [38,39,40]. To avoid unnecessary descriptions of previously published work [38,39,40], briefly, the combustion was regulated by adjusting the air flow reaching the combustion chamber (here, a steel chimney) using a flow meter/controller (Fisher Scientific 11998014, Schwerte, Germany). The flow controller was set to maintain an air flow of 0.0035 m^3^/min, which caused the deposition of ~15 µm thick soot with quasisquare morphology, root mean square roughness of ~100 nm, inherent mechanical robustness, and superhydrophobicity towards water-based liquids [41,42]. 

### 3.3. QCM Based System for Human Spermatozoa Quality Assessment

The functionality of as prepared and carefully stored spermatozoa was additionally studied by means of a fully-automated QCM 200 measurement system (SRS, San Diego, CA, USA) in order to verify its applicability and effectiveness for semen quality analysis. The QCM 200 package includes a digital controller with a built-in frequency counter and ohmmeter, to whose front panel are connected a sensor oscillator SRS 25 and a quartz crystal holder via RJ-45 and BNC connectors. Once the system is switched on and a 5 MHz QCM with active gold electrode diameter of *d* = 12 mm is placed in the holder, an alphanumeric LED display starts showing the real-time changes of the series resonance frequency *f_r_*, dynamic resistance *R*, and displaced mass *m* per certain gate time (1 s in this study). These continuous readings are transferred and recorded instantly on a personal computer, connected to the digital controller through RS-232 communication ports, using a specially designed LabView Stand Alone computer software (SRS, San Diego, CA, USA). 

Prior to the experiments, each part of the instrument was thoroughly sterilized with ethanol (70%, Sigma-Aldrich, St. Louis, MO, USA) and two uncoated and soot coated 5 MHz QCMs were placed one at a time in the crystal holder. The uncoated sensor was used as reference with an aim to correlate our results with those reported in refs. [24,25,26] and identify the major advantages of the superhydrophobic device configuration compared to the hydrophilic one (blank quartz crystal). Upon stabilization of the signal in air (Δ*f* ±1 Hz), ~1 mL Global for fertilization was added on the sensor surface and the upper part of the holder was covered with a sterilized beaker, minimizing/preventing the liquid evaporation. Afterwards, the ten-turn dial of the digital controller was manually set to null the static capacitance *C*_0_ and obtain an error-free liquid phase QCM response in time. Afterwards, ~10 µL droplets of human serum albumin (HSA) and motile spermatozoa with initial concentrations of 1–10–100 mg/mL and 100,000–1,000,000–10,000,000 units/mL, respectively, were consecutively added in the buffer on every 40 min (2400 s) via pipette. Based on the droplet volume, the final HSA and spermatozoa concentrations in Global were 10–100–1000 µg/mL (10–100–1000 ppm) and 1000–10,000–100,000 units/mL, accordingly. The above procedure was applied to both uncoated and soot coated QCMs in two independent measurement cycles, and in some of the experiments the motile spermatozoa were substituted with immotile (dead cells), which was expected to induce visible differences in the sensor response. All of the assays were conducted in an open lab at relatively constant room temperature and humidity (~25 °C, 50%). 

Several important reasons can justify the chosen experimental setup. First, the natural motion of spermatozoa is in upward direction, opposite to the gravitational forces and body fluids flow. Hence, although very practical, the availability of a swim cell does not completely compensate this upward motion, which may cause uncertainties in determining the sperm motility [26]. On the other hand, since the gold electrodes act as a receptor of spermatozoa [26], it is quite logical to assume that the actively motile cells will trigger different sensor response when compared to the immotile counterparts (e.g., through local/rapid motility-driven mass loading changes at the solid-liquid interface). Second, since the QCM 200 is capable of generating readable signal in highly viscous environments (R~5000 Ω), we can confidently anticipate meaningful data acquisition even at very noisy background, caused by the excitation of compressional waves or viscosity variations [26,28]. In fact, such a background can be utilized to evaluate the efficiency and resolution of the system. Third, we avoided mounting the apparatus in an incubator, because the resetting of C_0_ can be done only manually, which in turn will change the initial conditions set in the incubator (temperature, humidity, and CO_2_ content). Finally, we supplemented the Global medium with HSA, because the albumin is found in high concentrations in the human reproductive tract and sustains the vitality and motility of the gametes. 

## 4. Results and Discussion

### 4.1. Structure and Morphology of the Soot Coatings

Figure 1 represents the typical morphological and structural features of the water repellent soot coatings utilized for human spermatozoa analysis. They are composed of tightly connected and fused carbon nanoparticles, arranged into a fractal-like network of large solid aggregates (>200 nm) separated by micro- and nanoscale pores. Such a surface profile has recently been reported in many research articles of one of the current authors [23,38,39,40,41,42,43] and it corresponds to inherently robust superhydrophobic soot, composed of carbon and oxygen, with high static contact angle, low contact angle hysteresis (152–156°; 0.5–1°), and droplet bouncing behavior towards water [23,39,41,42,43,44]. 

### 4.2. Semen Quality Analysis Using Uncoated 5 MHz QCMs

The first part of this assay includes real-time observation and recording of the sensor signal of an uncoated 5 MHz QCM immersed in a composite liquid medium of Global, HSA, and immotile male gametes (MG), as shown in Figure 2. 

According to the experimental procedure in Section 3.3, analytes with increasing concentration (in Figure 2, HSA and immotile MG) are consecutively added in the buffer solution on every 2400 s, i.e., at 2400, 4800, 7200, 9600, 12,000, and 14,400 s. The careful examination of graphic dependences in Figure 2a–d reveals fair repeatability in the resonance frequency and mass displacement trends with an overall readable shifts within Δ*f* and Δ*m* ~125 Hz and 3 µg/cm^2^, respectively. These values are obtained by following the baseline’s shape rather than the maximum frequency and mass deviations, because of the appearance of sharp peaks with high and low intensity, compromising the accurate extraction of the useful sensor signal. Such sharp peaks may have two origins, namely the excitation of compressional waves resulting from non-uniform particle motion on the quartz surface [26,28] or random viscosity alterations within the shear wave penetration depth *δ*, since both HSA and human spermatozoa have ~5–9 times higher viscosity compared to that of aqueous solutions like Global [45,46]. Surprisingly, the increased mass loading on the sensor surface, due to the attachment of HSA and immotile MG, does not lead to gradual frequency decrease and mass displacement increase (as it should normally be). Furthermore, during the experiments, the resistance’s background is very noisy and does not provide a meaningful data threshold for separation of the protein and spermatozoa effects. Hence, the QCM’s detection mechanism for an uncoated sensor surface seems to be mainly governed by the viscosity of the buffer solution. Although speculative, this statement has its significance and will be discussed thoroughly later in this article. 

Other reason for the unexpected trends in Figure 2 might be related to interactions of HSA with the buffer’s ingredients (see Table 1), leading to impeded biomass adhesion. To study this possibility, we set an uncoated sensor in Global medium containing solely immotile or motile spermatozoa and juxtaposed its resonance behavior, as illustrated in Figure 3. 

Interestingly, the absence of HSA does not alter the frequency and mass displacement profiles for the case of immotile MG and they match those in Figure 2. However, different sensor response is observed by replacing the immotile spermatozoa with motile. As seen in Figure 3c,d, all spectra are characterized with quasisinusoidal shape and ~2.5–3 times smaller frequency and mass deviations compared to the data in Figure 3a,b. Moreover, there is relatively good repeatability of the signal, unlike the immotile counterparts that induce vast drifts from measurement-to-measurement (see Appendix A). Provoked by the contradicting results, we decided to fragment the signal and exclusively analyze the part due to the Global itself, while in the meantime examining the used uncoated QCM under a fluorescence microscope, as summarized in Figure 4 and Figure 5. 

The graphs in Figure 4, along with the data in Appendix A, reveal low repeatability of the sensor response induced by the buffer. It is important to note that the trends shown in Figure 2 and Figure 3 reflect real biological interactions at the solid-liquid interface, which is evident by the surface attached spermatozoa. Based on the images in Figure 5, smaller amount of motile sperm cells adhere to the uncoated (gold electrode) surface compared to the immotile counterparts. This observation is in full agreement with the real-time resonance characteristics of the QCM in Figure 3, where the motile spermatozoa induce ~2.5–3 times less sensor signal, which is likely due to their natural affinity to overcome the gravity forces and “swim” across the buffer. 

As closing remarks for this part of experiments, the uncoated 5 MHz QCMs are capable of detecting the presence of HSA and sperm cells with various concentrations, although not in the conventional way (i.e., frequency downshifts proportional to the mass loading). Furthermore, the sensor can differentiate the immotile and motile human spermatozoa, but not with the required consistency and repeatability. Substantial disadvantage is the equalized response that is obtained in the presence or absence of HSA (see Figure 2 and Figure 3a,b), which hinders the unambiguous separation of the protein and sperm effects. 

### 4.3. Semen Quality Analysis Using Superhydrophobic Soot Coated 5 MHz QCMs

After performing the initial studies and outlining the major shortcomings of uncoated QCMs, we investigated the potential of superhydrophobic carbon soot coatings for semen quality analysis. Figure 6 shows the evolution of the sensor signal upon loading the surface of a soot coated 5 MHz QCM with Global containing HSA and immotile or motile male gametes. 

A quick overview of Figure 6 divulges the main benefits of having a superhydrophobic sensor surface. First, the sensor signal is extremely stable throughout the experiment and there is no noise whatsoever. Second, the fraction of spectra up to the 9600 s (just before adding the gametes) is repeatable. Third, in both cases (immotile and motile spermatozoa) the dynamic resistance *R* stays constant in time with negligible drift of ~1 Ω (see Appendix A).

Fourth, the increased concentration of immotile spermatozoa leads to a stable and linear frequency decrease of Δ*f* ~19 Hz, while the motile gametes give rise to ~9 Hz upward frequency offset (see the fractions in Figure 6a,c, after 9600 s). Whilst the frequency trend for the immotile sperm cells is quite equal in both measurement cycles with ~3 Hz deviation, the motile spermatozoa act differently in the second experiment and induce ~20 Hz downshift (see Appendix A). Such a result is not something abnormal, since the sperm samples are collected from healthy individuals with similar seminal kinetic characteristics, but diverse sperm vitality. In other words, although progressively motile, some male gametes may preserve their motility for a long time, but others might lose it literally in a few minutes [47]. 

Of course, the observed discrepancies could also be ascribed to the presence of HSA; therefore, we excluded the protein and performed another set of experiments, summarized in Figure 7. 

In the absence of HSA, the superhydrophobic QCM is capable of accurately differentiating the motile and immotile sperm cells by triggering frequency shifts Δ*f* ~30–60 Hz and substantially changing the signal trend (see Figure 7). Unlike the uncoated sensor, the soot coated one provides repeatability of the results both in terms of sensitivity (Δ*f*) and resonance behavior (see Appendix A). 

For the sake of completeness, we placed one soot coated glass slide for 24 h in a Petri dish filled with Global, HSA, and motile male gametes, and subsequently examined its surface through fluorescence microscopy, as shown in Figure 8.

Impressively, most of the coating’s surface is dry and completely free of gametes, while only a few narrow wet regions across the soot contain a limited number of attached cells. Apparently, the soot exhibits anti-bioadhesion/fouling properties towards human spermatozoa, except to bacteria and marine organisms, and cell attachments can occur solely due to partially collapsed non-wettability of the coating [23,48]. In all other cases of non-penetrating liquid, the cells (here spermatozoa) “float” across and/or touch the solid-liquid interface without irreversibly binding to the surface. Such a statement is supported by the reduction of series resistance *R* of the soot coated QCM, as previously documented for bacteria [48] and explicitly shown in Table 2.

The data in Table 2 reveal “decoupling” of the liquid-phase QCM response (see additionally Section 2), as reported for soot and other types of interface coatings [23,29,32,33,34,35], and demonstrate lack of sperm adhesion. The latter is confirmed via the absence of energy losses (positive and increased *R*, as in the case of uncoated QCM), normally arising when the liquid environment wets the sensing surface and/or the biomass is rigidly attached [22,23,48]. This opens a lot of opportunities for future work, including the insertion of specific spermatozoa receptors in the soot to facilitate the sperm attachment and separate in-situ the motile and immotile species from a composite seminal mixture.

### 4.4. Insights into the Detection of HSA and Human Spermatozoa Using Uncoated/Soot Coated 5 MHz QCMs 

The extraordinary results obtained during the experiments and duly described in Section 4.2 and Section 4.3, determine the necessity of providing logical explanation of the observed dependences. Instead of yielding proportional frequency decrease and mass displacement increase upon attachment of immotile spermatozoa or HSA, as shown elsewhere [24,25,26,49], the uncoated QCM exhibits initial downward frequency movement (upward for the mass) that further ascends to higher frequencies (see Figure 2a,c and Figure 3a). In addition, quasisinusoidal resonance characteristics are recorded in the presence of motile sperm cells (see Figure 3c,d).

The equation of Kanazawa and Gordon (Equation (1)) implies that the sensor performance of an uncoated QCM in liquids is mostly affected by the viscosity-density product [16]. Following this statement, the immersion of the uncoated QCM in Global induces viscosity-density regulated sensor signal, but since the buffer has 32 ingredients, non-uniform oscillations at the solid-liquid interface may cause the formation of cyclic spurious resonances [26,28]. Indeed, such cyclic resonances are clearly visible in Figure 2, Figure 3 and Figure 4, but their intensity and shape vary in time and among the individual measurements. We attribute these nonconformities to a complementary effect that is aroused by the partial oxidation of Global in the ambient environment, leading to differences in its pH and viscosity [50].

Expanding the above ratiocinations, upon insertion of 10 µL droplets of HSA and/or human spermatozoa in the buffer, the analytes start diffusing within its bulk until partially or fully reaching the acoustic wave penetration depth region *δ*. At this stage, the overall viscosity of that region increases from *η* to *η* + *n*, because both HSA and sperm cells possess higher viscosity than that of water-based liquids [45,46], and the QCM reacts by decreasing its resonance frequency (see Figure 2a,c and Figure 3a). Since part of the analyte adheres to the sensor surface, the quantity of remained unbound protein/sperm molecules within the liquid’s bulk decreases, as a result of which the viscosity of the *δ* region decreases proportionally with (*η* + *n*) − *p*, where *n* and *p* are parameters accounting the number of diffused and bonded analytes, respectively. Taking into account that molecule attachments/detachments occur rapidly in a short timeframe (assume 1 s), prior to thermodynamic equilibration of the system, the liquid’s viscosity changes multiple times, which accounts for the resonance frequency downward and upward fluctuations throughout the experiment. 

Our hypothesis is proven by the smooth and repeatable responses of the superhydrophobic soot coated QCM recorded in Global medium and HSA (see Appendix A). The thin layer of air, inherent for every superhydrophobic coating, creates a strong acoustic reflection plane at the three phase contact line, and despite the sensor being loaded with liquid, its resonance is dominated mainly by the quartz plate’s thickness [29]. A small fraction of wave energy is still transferred to the liquid via the surface features, yielding diminished viscosity driven dissipation and much lower sensor signal than the theoretical predictions of Kanazawa and Gordon (see Table 2) [23,29,32,33,34,35]. It is pertinent to mention that the soot coated sensor is not affected/wetted by the buffer and prior to adding the analytes, the frequency shift is ~0 Hz (see Appendix A). 

Of particular interest are the totally different responses of the superhydrophobic QCMs towards immotile and motile gametes in the presence or absence of HSA (see Figure 6 and Figure 7). Regardless of the protein, the motile spermatozoa trigger positive frequency shifts, while the immotile cells lead to gradual frequency decrease. We attribute these results to the intrinsic ability of live sperm cells to float around the liquid rather than precipitating at the solid-liquid interface, which in our opinion is the case for immotile gametes. Furthermore, the soot coated QCM is loaded with sperm samples from healthy donors, whose seminal characteristics are not strictly identical i.e., although progressively motile, the percent of spermatozoa belonging to category *a* or *b* (see Section 3.1) varies within the individuals. For instance, ~60 Hz frequency upshift in Figure 7c is triggered by 42% of sperm cells category *a* and 19% category *b*. In the second measurement, the frequency trend is repeatable (see Appendix A), but the lower signal of ~40 Hz is attributed to the decreased amount of spermatozoa category *a* (11%, while *b* ~23%). In contrast, when the semen quality is relatively equal (e.g., 33–48% category *a* + *b*), the sensor response is quite repeatable, even for an uncoated QCM (see Appendix A). Last but not least, a few variations in the resonance behavior of the soot coated QCM after adding 10,000 units of motile MG could be correlated with the sperm vitality. When it rapidly degrades (for 4800 s in our case), the cells deteriorate their “swimming capabilities” across the liquid, leading to slight frequency downshift, while the frequency upshift in the second assay indicates preserved vitality (see the frequency spectrum in Appendix A at 4800 s). Some of our considerations are speculative, but promising for future clinical tests with precisely set initial conditions (e.g., spermatozoa with well-defined vitality), since the soot coated QCM seems to detect the sperm functionality in two ways: quantitatively and via entirely opposite sensor signal trends. Finally, the soot interacts with aqueous solutions primarily through hydrogen bonds [39,40,51], so the partial collapse of its superhydrophobicity and subsequent attachment of spermatozoa after 24 h residence in the liquid environment (see Figure 8) can be used as proof-of-concept that small changes in the material’s chemistry (by manipulating the air flow reaching the combustion chamber [38,39,40,41,42,43]) will engender mitigated or intensified solid-liquid interactions and different binding potential of the gametes i.e., tunable sensitivity and detection limit of the sensor [36].

## 5. Conclusions

The present article discussed in detail the efficiency of QCM as an analytical tool for in vitro fertilization practice, in particular, human spermatozoa quality assessment. It was found that the uncoated quartz crystals are effective in sensing in-situ various concentrations of biological agents, such as HSA and male gametes (up to 10 ppm and 1 ppb, respectively), but they fail to ensure the required repeatability of the signal or suffer from exorbitant noise levels, impeding the precise fragmentation of the protein and sperm effects. In contrast, the superhydrophobic soot coated QCM induced noise-free sensor responses, where the specific characteristics of the analytes (motility and vitality) were distinctly segregated. These findings demonstrate the potential applicability of the superhydrophobic QCM for in-situ sperm quality assessment of patients with oligozoospermia (sperm concentrations below 100,000 units/mL) or ones who need sperm extraction from the testicles and epididymis.

## Figures and Tables

**Figure 1 sensors-19-00123-f001:**
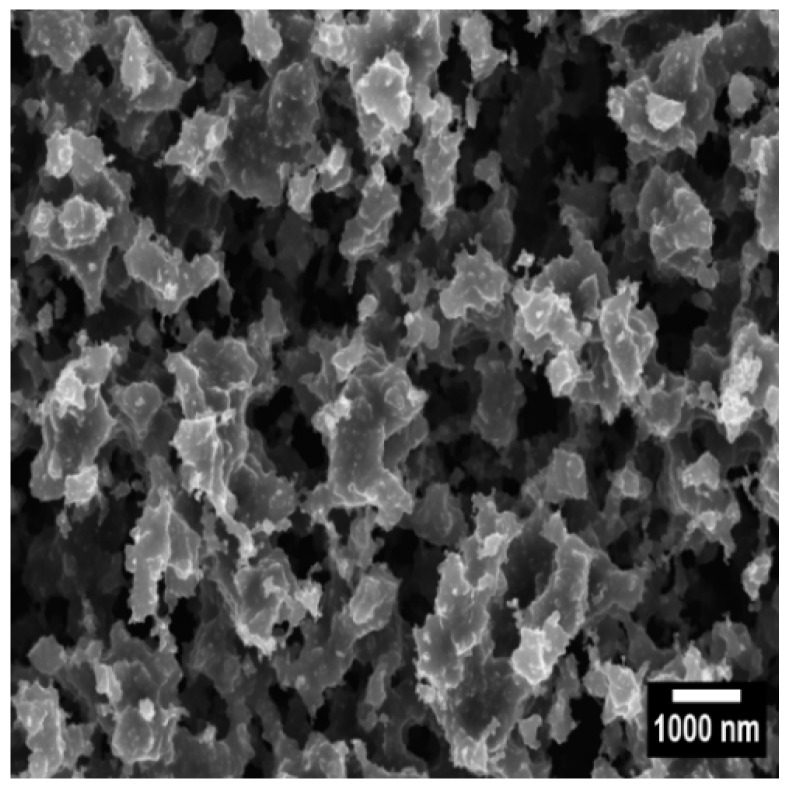
High-resolution scanning electron micrograph of the soot deposited during the controlled combustion of rapeseed oil at an air flow rate of 0.0035 m^3^/min.

**Figure 2 sensors-19-00123-f002:**
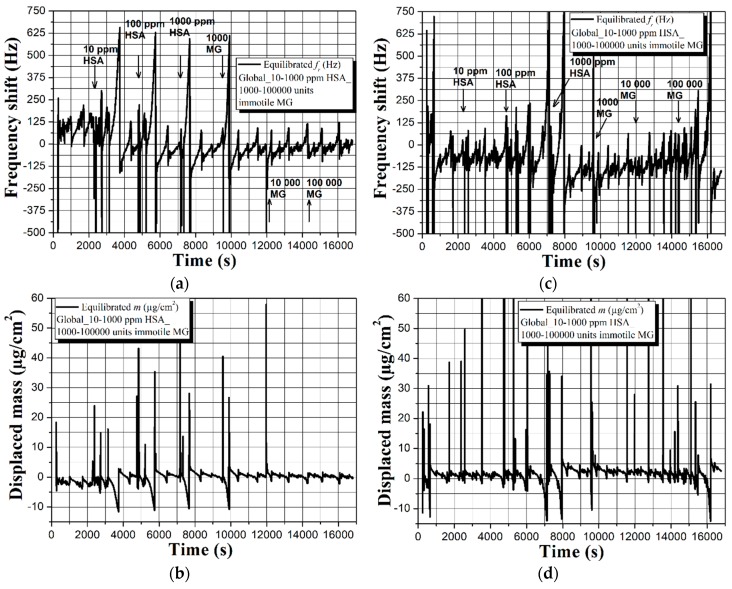
Real-time sensor response of an uncoated 5 MHz quartz crystal microbalance (QCM) to human serum albumin (HSA) and immotile human spermatozoa with gradually increasing concentration and its repeatability in two independent measurement cycles (**a**–**d**). All graphs reflect actual biomass effects and/or possible viscosity-density changes after equilibration of the baselines in Global medium.

**Figure 3 sensors-19-00123-f003:**
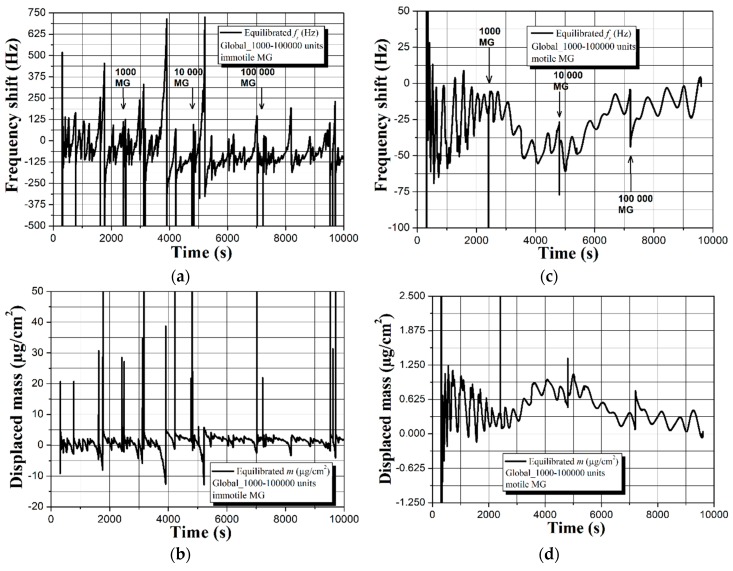
Real-time sensor response of an uncoated 5 MHz QCM to (**a**,**b**) immotile and (**c**,**d**) progressively motile human spermatozoa with gradually increasing concentration. All graphs reflect actual biomass effects and/or possible viscosity-density changes after equilibration of the baselines in Global medium.

**Figure 4 sensors-19-00123-f004:**
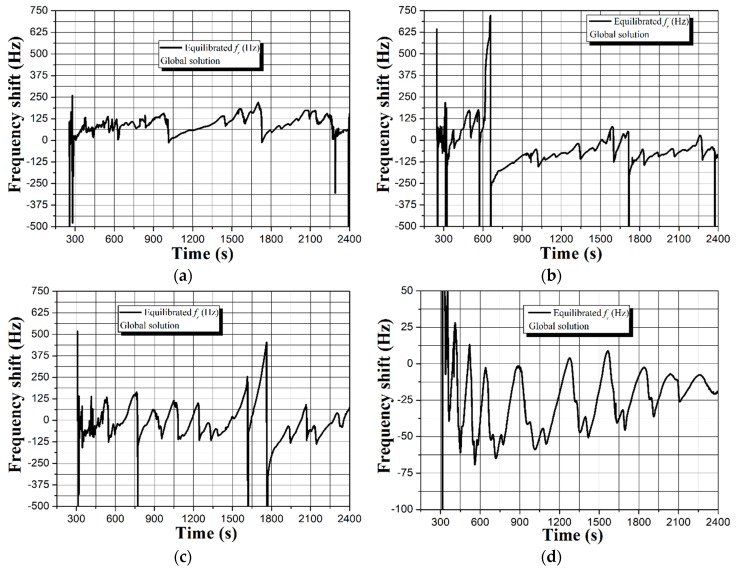
Fragmented frequency shifts for an uncoated 5 MHz QCM before adding (**a**) HSA and immotile spermatozoa at 1st measurement (part of Figure 2a), (**b**) HSA and immotile spermatozoa at 2nd measurement (part of Figure 2c), (**c**) immotile spermatozoa at 1st measurement (part of Figure 3a), and (**d**) motile spermatozoa at 1st measurement (part of Figure 3c). The full spectral profiles, i.e., those with mass displacement and resistance shifts can be found in Appendix A.

**Figure 5 sensors-19-00123-f005:**
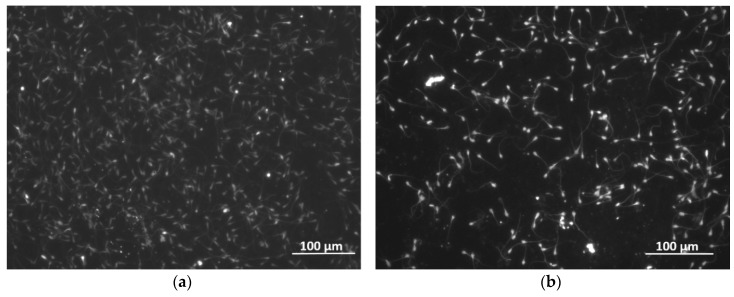
Fluorescence microscopy images of the uncoated sensor surface after the experiments with (**a**) immotile and (**b**) motile human spermatozoa.

**Figure 6 sensors-19-00123-f006:**
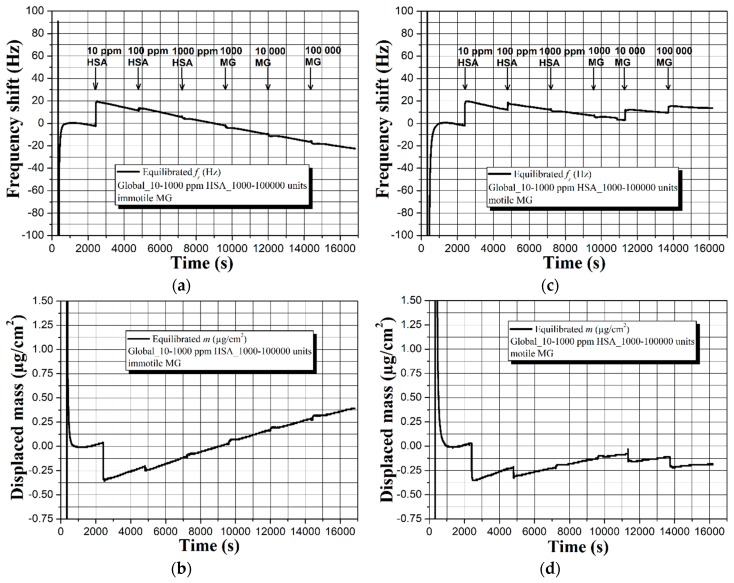
Real-time sensor response of a soot coated 5 MHz QCM to (**a**,**b**) HSA and immotile and (**c**,**d**) HSA and progressively motile human spermatozoa with gradually increasing concentration. All graphs reflect actual biomass effects after equilibration of the baselines in Global medium.

**Figure 7 sensors-19-00123-f007:**
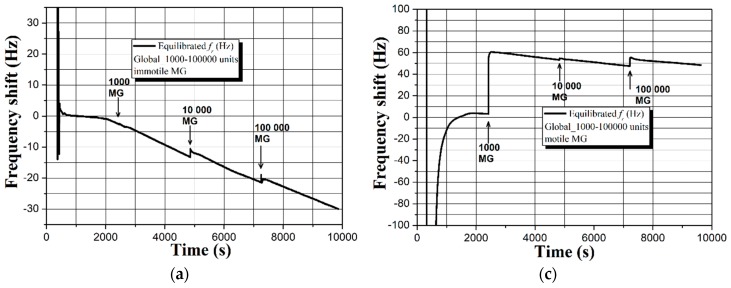
Real-time sensor response of a soot coated 5 MHz QCM to (**a**,**b**) immotile and (**c**,**d**) progressively motile human spermatozoa with gradually increasing concentration. All graphs reflect actual biomass effects after equilibration of the baselines in Global medium.

**Figure 8 sensors-19-00123-f008:**
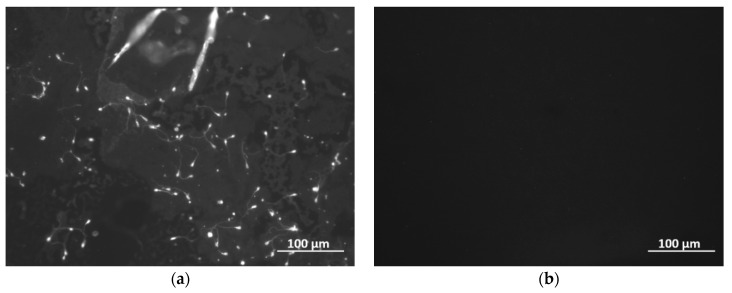
Fluorescence microscopy images of the (**a**) wet and (**b**) dry parts of soot coated glass slide after removal of the specimen from Global medium with HSA and motile human spermatozoa.

**Table 1 sensors-19-00123-t001:** Chemical composition of Global for fertilization medium.

Sodium Chloride	Sodium Pyruvate	Potassium Chloride	EDTA
l-Arginine	l-Threonine	l-Alanine	l-Cystine
l-Asparagine	l-Histidine	l-Aspartic Acid	l-Tyrosine
l-Glutamic Acid	l-Leucine	Potassium Phosphate	l-Valine
l-Phenylalanine	l-Methionine	Glycyl-l-Glutamine	l-Serine
l-Lysine	l-Proline	Sodium Bicarbonate	Glycine
l-Tryptophan	l-Isoleucine	Gentamicin Sulfate	Phenol Red
Calcium Chloride	Sodium Lactate	Magnesium Sulfate	Glucose

**Table 2 sensors-19-00123-t002:** Resonance frequency and series resistance values of uncoated and soot coated 5 MHz QCMs in air and after immersion in Global medium containing human spermatozoa.

Type of QCM	*f_air_* (MHz)	*R_air_* (Ω)	*f_Global_* (MHz)	*R_Global_* (Ω)	Δ*f* (Hz)	Δ*R* (Ω)
**Uncoated**	5.057125	29	5.055989	315	−1136	286
	5.055699	29	5.054717	314	−982	285
**Soot coated**	5.140457	185	5.140344	173	−113	−12
	5.140462	185	5.140312	171	−149	−14

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
