# Peer review of "Superhydrophobic Soot Coated Quartz Crystal Microbalances: A Novel Platform for Human Spermatozoa Quality Assessment"

_sensors, 2019, doi:10.3390/s19010123_

Round 1
Reviewer 1 Report
Esmeryan et al. reported that QCM device coated with soot, which produced superhydrophobic surface, had a potential to be used as a human spermatozoa quality assessment. They compared the assessment performance of gold thin film coated and soot coated QCM device. The gold coated device showed very noisy signals, in contrast, soot coated device could recognize the motility of spermatozoa. The paper is interesting and considered of value. I suggest that the paper is acceptable for publication in Sensors after major revision, because there are some claims as written below.
1. In section 3.2, the authors should describe the surface area of QCM device. There was no information about it.
2. Related to the above suggestion, data of mass loading (b, b’) in Fig. 2-4, and 6-8 should be deleted, because we can calculate the mass loading using data of frequency shift and Sauerbrey’s equation. There are so many graphs in the current manuscript.
3. Please insert the arrows at the drop operation with information of concentration on all graph in the QCM detection.
4. Please explain the reason why resistance decreased corresponding to the sample drop., for example, Fig. 6 (b’’) and 7 (b’’). I think that resistance will increased because viscosity increased by dropping the spermatozoa.
5. Average of the mobility of human spermatozoa should be analyzed and discussed in the revised manuscript.
6. Please check again that reference list was corrected. For example, I could not find ref. 29. It might be incorrect.
Author Response
We found the reviewers’ comments very helpful and constructive and we did our best to address them in the revised version of the manuscript. Revisions in the manuscript related with the comments and suggestions are highlighted in yellow. Our point-by-point responses to the reviewers’ comments are given below, each comment shown in italic and the modified text shown in regular font.
Referee 1:
Esmeryan et al. reported that QCM device coated with soot, which produced superhydrophobic surface, had a potential to be used as a human spermatozoa quality assessment. They compared the assessment performance of gold thin film coated and soot coated QCM device. The gold coated device showed very noisy signals, in contrast, soot coated device could recognize the motility of spermatozoa. The paper is interesting and considered of value. I suggest that the paper is acceptable for publication in Sensors after major revision, because there are some claims as written below.
1. In section 3.2, the authors should describe the surface area of QCM device. There was no information about it.
Response:
First, we would like to take the opportunity to thank the reviewer for his positive and thorough evaluation of the manuscript. The active electrode area (diameter) of commercially available gold electrode 5 MHz QCMs (SRS, USA) is 12 mm. Unfortunately, at the moment we are unable to determine the specific surface area of the soot, because we don’t have the equipment to do so (BET analysis). However, we know its film thickness (~15 µm) and root mean square roughness (~100 nm), and since this information was requested by the third reviewer, we have now provided it. Finally, the purchasing of equipment for BET analysis, as well as tribometer, is planned upon future funding acquisition (we have submitted a proposal under the framework Horizon 2020).
Change in Manuscript:
Information regarding the active electrode diameter, soot film thickness and roughness is available on Page 5, Lines 155-158 and 165 (sections 3.2-3.3).
2. Related to the above suggestion, data of mass loading (b, b’) in Fig. 2-4, and 6-8 should be deleted, because we can calculate the mass loading using data of frequency shift and Sauerbrey’s equation. There are so many graphs in the current manuscript.
Response:
We totally agree with the reviewer that the first draft of the manuscript contains a lot of graphs and their quantity needs to be reduced. Since this has also been pointed by reviewer 2, we carefully rethought the graph arrangements and came up with the following: Undoubtedly, the mass displacement can be calculated by knowing the frequency shift and using Sauerbrey’s equation. However, the main benefit of having these data is related to the fact that if the frequency shifts are triggered by actual biomass effects, the ∆f and ∆m signals must be diametrically opposite. In other words, the frequency downshifts must always be accompanied by mass displacement upshift and vice versa (i.e. frequency upshifts accompanied by mass displacement downshifts). This is clearly visible for the soot coated QCMs (Figures 6-7) and even for the very noise spectra shown in Figures 2-3. Therefore, having both ∆f and ∆m signals is very important for justification of the signals‘ validity i.e. that they are triggered by real biological effects rather than being an artefact due to the excessive noise. On the other hand, and as stated in the initial version of the manuscript, the series resistance’s background for uncoated QCMs is very noisy and does not provide meaningful data threshold for extracting useful information (see Page 8, Lines 236-237), while for soot coated QCMs the resistance is constant throughout the experiments (max 1 Ω deviation). In turn, and taking into account the reviewer’s useful suggestion, we decided to exclude the resistance shift graphs rather than the mass displacement ones.
For the convenience of all reviewers and editorial office, we show here the benefit of having ∆f and ∆m signals rather than ∆f and ∆R ones (resistance shifts are readily available in the supporting information).
a) | b) |
Figure A: One can clearly see the perfect agreement between the resonance frequency and mass displacement signals.
a) | b) |
Figure B: An independent reader may ask him/herself, how they can be sure the frequency shifts are due to the spermatozoa, since for the case of superhydrophobic QCMs the resistance shifts are negligible? (see also refs. 23, 29, 32-35 in the revised manuscript).
a) | b) |
Figure C: Although the signal generated by the uncoated QCM is very noisy, the frequency downshift up to 8000 s is supported by a displaced mass upshift at the same timeframe.
a) | b) |
Figure D: Low correlation between the frequency and resistance shifts.
Change in Manuscript:
According to the reviewer’s great recommendation, we have deleted 8 graphs and in our opinion the manuscript is streamlined.
3. Please insert the arrows at the drop operation with information of concentration on all graph in the QCM detection.
Response:
We are very grateful to the reviewer for this meaningful suggestion. Arrows indicating the type of analyte (HSA or MG) and its concentration have been inserted in the frequency shift graphs. We do not insert arrows for the mass displacement shift in order to avoid overloading of the graphs, but also because ∆f and ∆m trends are diametrically opposite. Thus, an independent reader can easily track the signals associated with the droplet (analyte) deposition.
Change in Manuscript:
Figures 2a, 2c, 3a, 3c, 6a, 6c, 7a and 7c have been revised and are now available on Pages 7-8, 11-12 in the revised manuscript.
4. Please explain the reason why resistance decreased corresponding to the sample drop. For example, Fig. 6 (b’’) and 7 (b’’). I think that resistance will increased because viscosity increased by dropping the spermatozoa.
Response:
We thank you very much for this question. For us, the reduction in series resistance is due to the “decoupling” of the liquid-phase sensor response and the way the sperm cells interact with the soot coated QCM. Similar result is reported in a previous article of Dr. Esmeryan, where a soot coated QCM possesses diminished sensor signal and negative resistance shift upon loading the surface with Pseudomonas putida (please see ref. 48 in the revised manuscript).
For your convenience, we attach a few graphs, which clearly support our considerations.
a) | b) |
Figure E: Sensor signal of soot coated QCM, when its surface is partially wetted by nutrient medium containing Pseudomonas putida. The graphs are taken from ref. 48 and used only for illustrative purposes.
a) | b) |
Figure F: Sensor signal of soot coated QCM, when its surface stays non-wetted upon contact with saline containing Pseudomonas putida. The graphs are taken from ref. 48 and used only for illustrative purposes.
Change in Manuscript:
Thorough explanation of the reasons for the observed negative trends of the series resistance is available on Page 13, Lines 316-331 in the revised manuscript.
5. Average of the mobility of human spermatozoa should be analyzed and discussed in the revised manuscript.
Response:
The human spermatozoa used in our study belong to category a+b, according to the standards of WHO. As such, their motility is as follows: category a - >25 µm/s and category b = 5-25 µm/s.
Change in Manuscript:
Information regarding the curvilinear velocity of the sperm samples is available on Page 4, Lines 143-144. This information is used also for interpretation of the signals generated by the soot coated QCM (please see section 4.4 in the revised manuscript).
6. Please check again that reference list was corrected. For example, I could not find ref. 29. It might be incorrect.
Response:
Thank you very much for this remark. Prior to submitting the manuscript, we have checked all references in the list for their correctness. Reference 29 is a conference paper published by prof. Glen McHale et.al. Here, we provide a link to this paper, but for your convenience, we will also upload a pdf file of this article. https://ieeexplore.ieee.org/document/4623089

Reviewer 2 Report
Eq. 1 by KANAZAWA does not support the result showing a positive shift of the frequency. Please explain.
The mobility of the individual human spermatozoom is statistically averaged. Please explain the cyclis change of the frequency shift.
The mobility must be observed under the microscope eyeshight. Please explain the relation between the mobility of the individual spermatozoom and the frequency shifu.
There are excessive amount of frequency shift data. Please explain the individual difference or please show carefully screened data.
Author Response
Referee 2:
1. Eq. 1 by KANAZAWA does not support the result showing a positive shift of the frequency. Please explain.
Response:
The reviewer is absolutely correct, but basically this mismatch between the equation of Kanazawa-Gordon and our results is one of the main novelties of the manuscript. What we mean: Normally, when the sensor surface is hydrophilic, the liquid loading induces resonance frequency downshift (~715 Hz for DI water) and series resistance increase (~350 Ω for DI water). However, and according to the four-layer theory (please see section 2), when the sensing surface is superhydrophobic, the inherent thin air film (plastron) localizes the wave oscillations mainly in the crystal’s bulk (this is due to huge mismatch between the acoustic impedances of solids and air). As a result, the amount of wave energy transferred from the crystal to the liquid and dissipated by it, is much lower compared to the case of hydrophilic sensor surface (where Eqs. 1 and 2 describe the resonance behavior). Then, the viscosity-driven dissipation is diminished and the sensor signal is much lower compared to the predictions of Kanazawa-Gordon. Now, in such a situation, where the influence of liquid’s viscosity is almost entirely suppressed, one of the ways of having sensor signal upon adding analytes is via their precipitation at the solid-liquid interface. Please note that we talk about precipitation, but not binding. In other words, the analytes do not firmly attach to the solid, as the superhydrophobic surface triggers large water contact angle (>150 °), which increases the separation distance between the solid surface and the target analyte. As a result, the kinetic barrier that the analyte needs to overcome in order to attach to the solid is much higher than the case of hydrophilic or hydrophobic surface. For more information on this topic, you may check ref. 48.
Saying that, the dead sperm cells (immotile) diffuse into the liquid and precipitate at the interface on gravimetric principle. In turn, the soot coated QCM induces visible frequency downshift of ~30 Hz (see Figure 7a). In contrary, and here is the main novelty of our manuscript, the progressively motile spermatozoa overcome the gravity forces, as this is their natural affinity (see also section 3.3) and this leads to positive frequency shift. So, exactly this non-trivial sensor response towards motile spermatozoa, along with the noise-free continuous signal, make the soot coated QCMs strong candidates for various in-vitro clinical analyses.
Change in Manuscript:
We are very grateful for the above remark, so the part in section 4.4 considering the frequency response of the superhydrophobic QCM was thoroughly rewritten. We hope that in its current revised form, the manuscript meets the high publication standards of MDPI Sensors.
2. The mobility of the individual human spermatozoom is statistically averaged. Please explain the cyclis change of the frequency shift.
Response:
The cyclic change of the frequency response, registered for the uncoated QCM (see Figures 2-4), is another very interesting aspect of the research. For hydrophilic surface such as the uncoated gold electrode QCM, the liquid-phase sensor response is mainly governed by the Kanazawa-Gordon equation. Taking into account the fact that Global for fertilization is water-based liquid with more than 20 ingredients, non-uniform oscillations or random viscosity alterations may occur and cause cyclic spurious resonances (see refs. 26, 28). It is very important to mention that the uncoated QCM senses liquid interactions only within the shear wave penetration depth δ ~250 nm (for water-based liquids and resonance frequency ~5 MHz).
So, when a ~10 µL is deposited in the buffer via pipette, the analyte starts diffusing into the bulk (unless it is motile spermatozoa, which can also float around without reaching the surface) and at some point reaches the shear wave penetration depth region. During the stage of analyte diffusion, the entire system is dynamically unstable and it is quite logical to expect non-uniform oscillations, hence, cyclic spurious resonances. Once the analyte reaches the shear wave penetration depth δ, the overall viscosity of that region increases, because both HSA and MG have higher viscosity than that of water (see refs. 45, 46). As a result, the uncoated QCM reacts to the increased viscosity and decreases its resonance frequency (supported with mass displacement increase). However, on the next stage of analyte binding, the total amount of analyte molecules within δ decreases. Why? Because some of them bind to the surface, so they no longer belong to the liquid δ region. Hence, and remembering that for uncoated QCMs the sensor response is mainly governed by the viscosity-density product of the liquid, the proportional decrease of analytes in δ causes local viscosity decrease and in turn resonance frequency increase. Now, knowing the binding events are reversible, prior to thermodynamic equilibration of the system, one can easily assume that the multiple molecule attachments-detachments will cause multiple viscosity changes within δ. Then, the resonance frequency will decrease and increase multiple times until at some point it stabilizes and steadily increases, because most of the molecules are already irreversibly attached (see the part of frequency spectrum in Figure 2a-2c after 10000 s.
Of course, we might be wrong in our hypotheses, but we do believe in the correct interpretation due to the lack of multiple frequency fluctuations upon coating the QCM surface with soot i.e. after minimizing substantially the influence of liquid’s viscosity on the sensor response.
Change in Manuscript:
We thank the reviewer for this very important remark and we have included one short and streamlined explanation of the observed cyclic frequency behavior in section 4.4, Page 14, Lines 346-357.
3. The mobility must be observed under the microscope eyeshight. Please explain the relation between the mobility of the individual spermatozoom and the frequency shifu.
Response:
This suggestion is very relevant and is also pointed by reviewer 3. The mobility was observed under microscope, this is how we separated the motile and immotile spermatozoa, and incubated only those with progressive mobility and curvilinear velocity of category a-b. Since two independent experts are rising the same question, we did our best to modify section 4.4 and provide quantitative explanation of the results.
Change in Manuscript:
Quantitative interpretation of the sensor signal is now available on Pages 14-15, Lines 370-391 in the revised manuscript.
4. There are excessive amount of frequency shift data. Please explain the individual difference or please show carefully screened data.
Response:
We agree with the reviewer. The revised version of the manuscript is streamlined and 8 graphs have been deleted. The individual difference is explained with a caution below each figure (for the case, where 2 or 3 figures look quite identical. Actually, as we comment on Page 10, Lines 270-275, the main disadvantage of the uncoated QCM is the equalized response obtained in the presence or absence of HSA, which hinders the unambiguous separation of the protein and sperm effects) or text within the figure.

Reviewer 3 Report
The idea of using soot-coating to improve the sensor is interesting, and the results did show great improvement in terms of noise. However, to use this design as a sensor for spermatozoa detection is seems to be hard to implement. The frequency shift seems to be irrelevant to the concentration. Also, it is not clear how to distinguish motile and immotile sample. It seems not clear to me how to use this sensor and read value (or a value range) of frequency shift, and correlate it to the "quality" (in term of concentration and mobility) of the spermatozoa. The authors need to elaborate more about this.
The authors need to provide more details about the soot coating. What is the thickness and roughness of the coating? Please also provide R or Q value of the quartz before and after coating, both under air and immersed in DI water.
Author Response
Referee 3
1. The idea of using soot-coating to improve the sensor is interesting, and the results did show great improvement in terms of noise. However, to use this design as a sensor for spermatozoa detection is seems to be hard to implement. The frequency shift seems to be irrelevant to the concentration. Also, it is not clear how to distinguish motile and immotile sample. It seems not clear to me how to use this sensor and read value (or a value range) of frequency shift, and correlate it to the "quality" (in term of concentration and mobility) of the spermatozoa. The authors need to elaborate more about this.
Response:
We would like to thank the reviewer for his/her positive, but critical and constructive comments. First, we would like to emphasize that our research is pioneering and as such we only claim novel platform for human spermatozoa quality assessment with potential applicability of superhydrophobic carbon soot coating as an interface material (please see Page 2, Lines 69-77). Whether soot coated QCMs may or may not be used in commercial (daily) clinical tests will depend on the extent to which this research will be developed. The present manuscript is the first step and we look forward to conducting more and more experiments, and clarifying every single detail related to the sensor response. We agree that at the moment the frequency shifts are somehow irrelevant to the concentration i.e. it seems the QCMs trigger equal shift regardless of the concentration (1000, 10 000 or 100 000 units). However, undisputedly, the sensors feel the presence of MG even at very low concentrations i.e. the device changes its working parameters once MG are inserted in the system and reach the shear wave penetration depth region. From that point-of-view, the claim “The obtained results reveal strong potential of the superhydrophobic QCM for future inclusion in diverse laboratory analyses closely related to the in vitro fertilization procedures, with a final aim of gaining practical approaches for diagnoses and selection of male gametes.” is scientifically correct.
The soot coated QCM, and even the uncoated one, distinguish the motile and immotile spermatozoa by inducing fundamentally different signal trends. In the case of uncoated sensor, the motile spermatozoa lead to quasi-sinusoidal signal, which is repeatable in two independent measurement cycles (see Figure S1 in the SI). On the other hand, the superhydrophobic QCM induces positive frequency shifts, when motile sperm cells are added in the system, while reacts in a conventional way upon insertion of dead MG (resonance frequency downshift, supported by mass displacement upshift – please see Figure 7).
However, the reviewer is undoubtedly correct about the “read value of frequency” and its relation to the semen quality. This part was missing in the first draft of the manuscript. Therefore, we carefully considered this suggestion, checked the seminograms of each donor and came up with some important conclusions, which in our opinion helped a lot to enhance the overall scientific quality of the manuscript.
Change in Manuscript:
Changes in the text, related with this reviewer’s concerns, are now available on Pages 14-15, Lines 370-391 in the revised manuscript.
2. The authors need to provide more details about the soot coating. What is the thickness and roughness of the coating? Please also provide R or Q value of the quartz before and after coating, both under air and immersed in DI water.
Response:
Thank you very much for this suggestion. The film thickness is approximately 15 microns, while the root mean square roughness is ~100 nm. We provided a new table with f, R, ∆f and ∆R values prior to and after soot deposition, both in air and upon immersion in Global with spermatozoa. This table helps us to support some of our conclusions, as it can be seen on Page 13. Regrettably, we cannot provide any data about the quality factor Q, because its calculation is possible if knowing C or L. These two parameters can be measured via PI-scheme, but for that purpose the 5 MHz QCMs need to be mounted in specially designed metal constructions. Acoustoelectronics Laboratory at ISSP, Sofia, Bulgaria has such constructions for small, 8 mm diameter, 16 MHz QCMs and their electrical parameters are readily measurable (please see Table 1 in Esmeryan et.al. A superhydrophobic quartz crystal microbalance based chemical sensor for NO2 detection, Bul. Chem. Commun. 47 (2015) 1039-1044). We are planning to measure C and L for 5 MHz QCMs in the near future.
Final comment:
Once again, we take the opportunity to sincerely acknowledge all three reviewers for their professional, critical and constructive comments. As pioneering work (human spermatozoa quality assessment via soot coated QCM), we understand that some of our hypotheses and conclusions may not be completely correct. This can be attributed to the lack of foundation to step upon (previously published data for sperm detection with superhydrophobic QCMs). Nevertheless, we strongly believe and hope the results presented here have significant scientific merit and substantially advance the field of QCM based biosensors.

Round 2
Reviewer 1 Report
The revised manuscript refrects our reviewers' comments. It can be accepted for publication.
Author Response
We would like to thank the reviewer for his/her support. Indeed, we also believe that the revised manuscript reflects in the best possible way all reviewers’ comments.

Reviewer 3 Report
The responses provided by the authors have answered most of my question, and the manuscript has been improved as well.
There is not doubt that the soot coating has greatly improved the noise. However, to claim this as a "sensor", the stability of the coating in the media is important. Regarding the discussion about frequency shift vs spermatozoa quality, to be more precise, my concern is that the fairly constant frequency decrease regardless of the concentration (in Figure 6 for immotile sample) indicates that the frequency shift might be caused by the soot coating swelling/wetting in the media. To rule this possibility out, the authors need to provide a controlled data of frequency vs time (up to 16000s) in the Global media without human spermatozoa. This will provide a more accurate interpretation of the data.
Author Response
1. The responses provided by the authors have answered most of my question, and the manuscript has been improved as well. There is not doubt that the soot coating has greatly improved the noise. However, to claim this as a "sensor", the stability of the coating in the media is important. Regarding the discussion about frequency shift vs spermatozoa quality, to be more precise, my concern is that the fairly constant frequency decrease regardless of the concentration (in Figure 6 for immotile sample) indicates that the frequency shift might be caused by the soot coating swelling/wetting in the media. To rule this possibility out, the authors need to provide a controlled data of frequency vs time (up to 16000s) in the Global media without human spermatozoa. This will provide a more accurate interpretation of the data.
Response:
We are very happy that the revised manuscript has answered most of the reviewer’s comments. His main concern whether the frequency shift for immotile gametes (“fairly constant frequency decrease regardless of the concentration”) is caused by wetting of the soot with the Global medium, has been addressed even in the previous version of the manuscript, but not in an explicit way.
Before starting the experiments, we have taken this possibility into account (soot delamination/collapsed non-wetting due to the liquid environment), therefore, each time the soot coated QCM was loaded with Global, we kept it for about 40 min prior to adding the analytes.
Figure S10 in the supporting information clearly shows constant frequency shifts of approximately 0 Hz, generated by the pure Global medium (no analyte). This has been confirmed in 8 independent measurement cycles i.e. 8 multiply to ~10 000 s results in 80 000 s or 22.2 hours continuous immersion of the coating in liquid medium, without and later with analytes (2 for HSA and immotile MG; 2 for HSA and motile MG; 2 for immotile MG/lack of HSA and 2 for motile MG/lack of HSA). In all these measurements, the signal in the first 2400 s (40 min) is only due to the Global and one may see that in 4 of 8 cases the frequency response is ~0 Hz, while in the other 4 cases the maximum recorded frequency shift is ~8 Hz. These data undisputedly confirm the preserved non-wetting of the soot coated QCM, since if the coating was wetted, the signal would be in the range of a few hundreds, even thousands Hz (you may see Figure Ea in the previous revision cover letter. There, the overall frequency shift upon partial wetting of the coating is far higher than 30 Hz. It is about 1500 Hz. One may say, yes, but part of it is due to the bacteria themselves and this is true, but at least ~715 Hz of the signal result from the liquid. The latter is the theoretically calculated value, using the Kanazawa-Gordon equation, for water-based liquids, when the sensor surface is no longer extremely water repellent.)
Furthermore, another evidence for the lack of wetting is that after each experiment with the soot coated QCM, its surface was completely dry and the resonance frequency was restoring its initial baseline value with very negligible changes within a few Hz (you may see the second row in Table 2 in the revised manuscript. Our statement is clearly confirmed there).
For the convenience of the reviewer and honorable editorial office, we add here some of the graphs from Figure S10.
a) | b) |
c) | d) |
Figure A: Frequency shift of the soot coated QCM in pure Global environment (no analytes) in four independent measurement cycles (a-d).
Finally, we strongly hope that this round of review will be the last one and the manuscript will be accepted for publication, as suggested by reviewer 1. We did our best to comment upon all critical aspects of the research. We are quite sure that a lot of question will be further raised by the scientific community, which in our opinion is excellent. One of the famous professors at the Institute of Electronics Bulgarian Academy of Sciences, prof. Petar Atanassov, says the following: “If one research is good, there are always a lot of questions to be raised, if the research is low quality, there is nothing to ask for”.
Change in Manuscript:
A sentence emphasizing the lack of wetting on the soot coated QCM is now available in section 4.4, Page 14, Lines 364-366 in the revised manuscript.
